# Dedifferentiation-Dependent Regeneration of the Biliary Ductal Epithelium in Response to Hepatic Injury in TFF1-Deficient Mice

**DOI:** 10.3390/cells14171323

**Published:** 2025-08-27

**Authors:** Taisuke Yamamoto, Junpei Yamaguchi, Toshio Kokuryo, Yukihiro Yokoyama, Takashi Mizuno, Shunsuke Onoe, Masaki Sunagawa, Taisuke Baba, Tomoki Ebata

**Affiliations:** Division of Surgical Oncology and Nagoya University Graduate School of Medicine, 65 Tsurumai-cho, Showa-ku, Nagoya 466-8550, Japan

**Keywords:** trefoil factor, biliary epithelial cells, hepatic progenitor cells

## Abstract

The mechanisms involved in the regeneration of biliary epithelial cells (BECs) after liver injury remain unclear. In this study, we employed KRT19Cre^ERT^/LSL-tdTomato (KT) mice and KT/TFF1KO mice to clarify the regeneration and cell fate of BECs via lineage tracing. Tamoxifen (TAM) was administered to the mice to label cytokeratin 19 (CK19)-positive BECs. The mice were subsequently fed a choline-deficient, ethionine-supplemented (CDE) diet for four weeks, after which the mouse livers were analyzed. Whereas the proportion of tdTomato^+^ cells in CK19-positive BECs decreased in the KT mice, it remained high in the KT/TFF1KO mice. Then, we analyzed hepatic progenitor cells (HPCs), the possible source of BECs. Although tdTomato-labeled HPCs were rarely found in the pretreatment mice, they were frequently found in the KT/TFF1KO mice after the CDE diet, suggesting the dedifferentiation of tdTomato-labeled BECs to HPCs. These results indicate not only that the loss of TFF1 accelerates the dedifferentiation of BECs into HPCs but also that HPCs are the source of BECs in TFF1KO mice. In addition, tdTomato-labeled HNF4α-positive hepatocytes were frequently found in the KT/TFF1KO mice, revealing the transdifferentiation of BECs to hepatocytes. The role of TFF1 as an inducer of biliary differentiation might be useful in the treatment of patients with hepatic or biliary dysfunction.

## 1. Introduction

The liver is a sophisticated regenerative organ that has a strong ability to recover after liver damage, including partial hepatectomy or toxin-induced injury, with reproduction of a damaged epithelium. Hepatocytes and cholangiocytes are the dominant epithelial cells of the liver, with hepatocytes accounting for approximately 80% and cholangiocytes accounting for 3% under homeostatic conditions [1]. While the origin of regenerative epithelial cells in the liver has been investigated and reported in mouse models, the role of hepatic progenitor cells (HPCs) remains controversial because the regenerative process differs depending on the mouse strain and model of liver injury [2,3].

Hepatocytes maintain low proliferative levels under homeostasis, whereas they show high proliferative activity after liver injury. Despite the early assumption that the regeneration of hepatocytes depends on the differentiation of HPCs, gene lineage tracing studies have shown that self-renewal of pre-existing mature hepatocytes primarily contributes to their regeneration [4,5]. More recently, however, investigations have revealed that not only HPCs but also cholangiocytes contribute to hepatocyte regeneration when the proliferative capacity of hepatocytes is impaired by chronic liver injury [6,7,8,9]. HPCs may also play a critical role in the regeneration of biliary epithelial cells (BECs) [10]. Extensive proliferation of BECs can typically be observed as a ductular reaction in chronic liver diseases such as nonalcoholic steatohepatitis [11,12], which can arise from HPCs; however, the precise role of HPCs in the regeneration of BECs remains controversial. In addition, the mechanisms involved in the differentiation of HPCs into BECs remain unclear, but the Wnt pathway accelerates the differentiation of HPCs into hepatocytes, and the Notch pathway accelerates the differentiation into BECs [13].

The trefoil factor family (TFF) comprises small proteins expressed mainly in the gastrointestinal mucosa. Among the three TFFs, TFF1 is abundantly expressed in the gastric mucosa and has various functions, including protection of the gastrointestinal mucosa, epithelial regeneration, and anti-inflammatory effects [14]. More recently, several reports revealed that TFF1 functions as a tumor suppressor in the gastrointestinal epithelium such that TFF1 deficiency is associated with the development and progression of gastric cancer and Barrett’s esophageal cancer [15,16,17]. TFF1 is also widely expressed in various organs, such as the brain, tracheal epithelium, lymph nodes, pancreatic ducts, and liver [18,19,20,21,22]. TFF1 is abundantly expressed in BECs in patients with chronic cholecystitis and intrahepatic lithiasis [23]. Mechanistically, we previously reported that the tumor-suppressive function of TFF1 in pancreatic and liver carcinogenesis involves the inhibition of Wnt pathway activation [24,25]. Overall, we hypothesized that TFF1 can regulate the regeneration of BECs and the differentiation of HPCs into BECs.

In this study, we employed a lineage-tracing mouse model of BECs and investigated the process of BEC regeneration after liver injury, with an emphasis on the role of TFF1 and the contribution of HPCs.

## 2. Materials and Methods

### 2.1. Animals

Krt19-Cre^ERT^ (catalog #026925) and LSL-tdTomato (catalog #007914) mice were purchased from the Jackson Laboratory (Bar Harbor, ME, USA). TFF1-knockout (TFF1KO; knockout-first allele: Tff1^tm1a (EUCOMM)Wtsi^) mice were purchased from the International Mouse Phenotyping Consortium. C57BL/6J mice were purchased from SLC Japan (Nagoya, Japan). These mice were bred to generate KT (Krt19-Cre^ERT^/LSL-tdTomato), KT/TFF1^+/−^, and KT/TFF1^−/−^ mice, and lineage tracing was performed.

At 2 months of age, 3 mg of tamoxifen (TAM) was administered orally for 3 consecutive days to label cytokeratin 19 (CK19)-positive BECs with tdTomato. These mice were defined as “Before treatment” [KT (*n* = 8), KT/TFF1^+/−^ (*n* = 8) and KT/TFF1^−/−^ (*n* = 8)]. After a one-week washout period, we induced three types of liver injury in the mice. (1) CDE model: Mice were fed a choline-deficient, ethionine-supplemented (CDE) diet for 4 weeks, after which their livers were harvested [KT (*n* = 11), KT/TFF1^+/−^ (*n* = 12) and KT/TFF1^−/−^ (*n* = 9)]. (2) CCl_4_ model: Mice were intraperitoneally injected with tetrachloride (CCl_4_; 2.5 mL/kg, 20% CCl_4_ dissolved in corn oil) twice a week for three months, after which their livers were harvested [KT (*n* = 13), KT/TFF1^+/−^ (*n* = 10) and KT/TFF1^−/−^ (*n* = 10)]. (3) In the BDL model, the common bile duct was ligated under anesthesia, and the liver was harvested one week after surgery [KT/WT (*n* = 12), KT/TFF1^+/−^ (*n* = 10) and KT/TFF1^−/−^ (*n* = 10)]. All animal experiments were conducted in accordance with the guidelines of the Institute for Laboratory Animal Research, Nagoya University Graduate School of Medicine.

### 2.2. Histology and Immunohistochemistry

The specimens were fixed for 24–48 h in 10% formalin/PBS. Histological analysis was performed on 4 μm paraffin-embedded sections that were stained with hematoxylin and eosin (HE). Immunohistochemistry (IHC) was performed using the antibodies listed in Table 1. First, the paraffin-embedded sections were deparaffinized and washed with PBS. For antigen retrieval, the slides were microwaved for 10 min at 100% power in pH 6 sodium citrate buffer [tdTomato, SRY-box transcription factor 9 (SOX9)] or pH 9 Dako Target Retrieval Solution [CK19, hepatocyte nuclear factor 4 alpha (HNF4α)]. After cooling to room temperature, the slides were placed in 10% H_2_O_2_ dissolved in methanol for 15 min to quench endogenous peroxidase activity. After being washed with PBS, the slides were filled with 10% goat serum blocking solution or bovine serum albumin for 10 min. Then, the slides were incubated with primary antibodies overnight at 4 °C and incubated with horseradish peroxidase secondary antibodies for 30 min at room temperature. A Dako REAL Envision Detection System Peroxidase/DAB+ was used for development, and the slides were counterstained with hematoxylin. Ten nonoverlapping fields of view (×100 microscopic field) were randomly selected via a Keyence BZ-X800 (Keyence, Osaka, Japan), evaluated with ImageJ (version 1.51j8) software from the National Institutes of Health (NIH), and cells were counted in a blinded manner.

### 2.3. Immunofluorescence

For double immunofluorescence staining [IF: CK19/tdTomato, alpha-fetoprotein (AFP)/tdTomato], cocktail antibodies were prepared as primary antibodies, and the slides were incubated overnight at 4 °C. Then, the slides were incubated for 45 min at room temperature with secondary cocktail antibodies (Alexa Fluor 555 and Alexa Fluor 647: Cell Signaling Technology, Danvers, MA, USA). The slides were washed three times with PBS and incubated with DAPI. Ten nonoverlapping fields of view (×400 microscopic field) around Glisson’s sheath were randomly selected and imaged via a Keyence BZ-9000 system (Keyence, Osaka, Japan), and the nuclei were visually counted.

Triple IF (CK19/SOX9/tdTomato, CK19/HNF4a/tdTomato) was performed via the Opal method with an Opal 3-Plex Manual Detection Kit (KIKOTECH, Osaka, Japan) according to the manufacturer’s protocol. Briefly, after treatment with the primary antibody (CK19) and secondary HRP-conjugated antibody, the slides were filled with Opal 570 Reagent and incubated for 20 min. After washing, the slides were microwaved for 1 min at 100% power to remove primary and secondary antibodies. The sections were subsequently microwaved for 15 min at 20% power in AR6 or AR9 buffer for antigen retrieval. After cooling to room temperature, the slides were incubated with secondary primary antibodies (against SOX9 or HNF4α) and with secondary horseradish peroxidase (HRP)-conjugated antibodies, after which they were incubated with Opal 690 Reagent for 20 min. The slides were treated with a third primary antibody (tdTomato) and secondary antibody and then treated with Opal 520 Reagent. Counterstaining with DAPI was performed, and 10 fields of view (×400 microscopic fields) were randomly selected and imaged.

### 2.4. Statistical Analysis

Continuous variables are presented as the means ± standard errors and were compared via Student’s *t* test. All the statistical analyses were performed via the Statistical Package for the Social Sciences (SPSS) version 28 (Chicago, IL, USA). A *p* value of <0.05 indicated a significant difference.

## 3. Results

### 3.1. Loss of TFF1 Resulted in the Proliferation of TdTomato-Labeled Cells

The histological phenotypes of the livers of the KT, KT/TFF1^+/−^ and KT/TFF1^−/−^ mice were compared via HE staining and IHC, which revealed no obvious differences. After TAM treatment (Figure 1A), we evaluated each genotype of mouse with IHC for CK19 and tdTomato (Figure 1B). CK19 and tdTomato were expressed not only in the bile duct epithelium with a glandular structure but also in the cells without it (arrows), confirming that CK19 was expressed even in micro BECs. The number of BECs was compared among the mice, revealing no differences in CK19 and tdTomato expression (Figure 1C,D). The mice with each genotype were subsequently fed a CDE diet for 4 weeks to evaluate bile duct regeneration during liver injury (Figure 1E–H); the CDE diet model is a mouse model of steatohepatitis in which hepatocytes and cholangiocytes are damaged. HE staining revealed fatty droplets and infiltration of inflammatory cells without apparent morphological differences between the genotypes. IHC revealed diffuse proliferation of CK19-expressing micro-BECs in all genotypes of mice, but the difference was not significant. In contrast, tdTomato-expressing cells were found more frequently in the KT/TFF1^−/−^ mice than in the KT mice. These results suggested that the tdTomato-labeled cells proliferated and contributed to the regeneration of BECs in the KT/TFF1^−/−^ mice.

### 3.2. The Labeling Efficacy Was Equivalent Among the Mice

To confirm the labeling efficacy of CK19-positive BECs with tdTomato after TAM administration, we performed double IF. To optimize effective dose of TAM, three to five times of TAM administration in KT mice was performed, confirming similar labeling efficacy (Appendix A). CK19- and tdTomato-positive cells were visually counted, revealing that approximately 45% of the CK19-positive BECs in the mice of each genotype were labeled with tdTomato, and the difference among the mice was not significant (Figure 2). This efficacy was comparable to that in a previous report [26] and indicated that TFF1 deficiency does not affect the labeling efficacy.

### 3.3. The Proportion of TdTomato-Positive BECs After CDE Varied Depending on TFF1 Status

To investigate the precise changes in cell number, we performed double IF of the CDE mouse model, and the numbers of CK19-positive and tdTomato-labeled BECs were visually determined (Figure 3A). While an increase in the number of CK19-positive BECs caused by CDE-induced injury was detected in each mouse (Figure 3B), with an approximately 5-fold increase in the KT mice and a 4-fold increase in the KT/TFF1^−/−^ mice, the total number of CK19-positive BECs in the KT mice was significantly greater than that in the KT/TFF1^+/−^ and KT/TFF1^−/−^ mice, indicating that the loss of TFF1 inhibited the regeneration of BECs in the CDE model.

If the proliferation of BECs is inhibited in TFF1-deficient mice, then the number of tdTomato-labeled BECs should be lower in KT/TFF1^−/−^ mice; however, many tdTomato-positive BECs were found in KT/TFF1^−/−^ mice, as noted above. To further explore biliary regeneration in KT/TFF1^−/−^ mice, we determined the numbers of CK19^+^tdTomato^-^ BECs and CK19^+^tdTomato^+^ BECs in each mouse. As a result, the number of CK19^+^tdTomato^-^ BECs was significantly lower and that of CK19^+^tdTomato^+^ BECs was significantly greater in the KT/TFF1^−/−^ mice (Figure 3C,D). We calculated the proportion of tdTomato-positive BECs and found it was significantly greater in the KT/TFF1^−/−^ mice (Figure 3E). Notably, this value was decreased by CDE treatment in the KT mice (approximately 44.4% to 26.5%) and remained at the same level in the KT/TFF1^−/−^ mice (approximately 44.4% to 45.9%). These results indicated that the regenerated BECs were predominantly tdTomato-negative cells in KT and tdTomato-positive cells in KT/TFF1^−/−^ mice.

### 3.4. Model of the Differentiation of HPCs and Biliary Regeneration

We then hypothesized that the regeneration of BECs was dependent on HPCs, especially in CDE models. If the regeneration of BECs was solely dependent on the self-proliferation of BECs, the proportion of tdTomato-labeled BECs should remain at the same level after regeneration (Figure 4A,B). What if the differentiation of HPCs contributes to the regeneration of BECs? Assuming that HPCs were not labeled with tdTomato, newly generated BECs from HPCs would be tdTomato-negative, resulting in a decrease in the number of tdTomato-positive BECs (Figure 4C). In contrast, BECs could have dedifferentiated into HPCs due to the loss of HPCs caused by severe liver damage. If this was the case, HPCs can be derived from tdTomato-labeled BECs, and there would be tdTomato-positive HPCs, which can differentiate again to supply tdTomato-positive BECs (Figure 4D). Hypothetically, tdTomato-positive HPCs can differentiate into tdTomato-positive hepatocytes, eventually resulting in the transdifferentiation of BECs to hepatocytes (Figure 4E).

### 3.5. TdTomato-Labeled BECs Dedifferentiate into HPCs Frequently in KT/TFF1^−/−^ Mice

Under the hypothesis that the origin of BECs in KT/TFF1^−/−^ mice is tdTomato-positive HPCs, we performed triple IF with CK19, tdTomato, and SOX9. Although SOX9 is a possible marker of HPCs, it is expressed even in mature BECs; thus, we defined SOX9^+^CK19^-^ cells as HPCs in this study (Figure 5A; arrows indicate HPCs, and arrowheads indicate mature BECs). HPCs were rarely labeled with tdTomato, and the total number of HPCs did not significantly differ among the mice before treatment (Figure 5B). In contrast, the number of HPCs increased, and tdTomato-positive HPCs appeared in the CDE-treated mice (Figure 5C,D). These results indicate that CK19^+^tdTomato^+^ BECs dedifferentiate into HPCs and that these cells are referred to as dedifferentiated HPCs (dHPCs). Interestingly, the number of dHPCs was significantly greater in the KT/TFF1^−/−^ mice (Figure 5E), suggesting that the loss of TFF1 resulted in the frequent dedifferentiation of BECs into HPCs and that the origin of regenerative BECs in KT/TFF1^−/−^ mice was tdTomato-labeled dHPCs.

### 3.6. Mathematical Model of HPC-Dependent Regeneration of BECs

To better understand the dynamic change in cell number, we aimed to establish a mathematical model of the regeneration of mature BECs with HPCs. Under the hypothesis that the regeneration of BECs depends on the proliferation of mature BECs and the differentiation of HPCs (Appendix A), the coefficients were set as follows: (a) the proliferation ratio of BECs, (b) the differentiation ratio of HPCs into BECs, (c) the proliferation ratio of HPCs, and (d) the death ratio (cellular apoptosis or necrosis due to liver injury) of all cells. If the proliferation rate of BECs depends on the deficiency in the number of cells, the dynamics of BECs can be described as$$a_{n}=k\left(\alpha -X_{n}\right)$$

Here, $a_{n}$ describes the proliferation ratio of BECs, and $X_{n}$ describes the number of BECs at the *n*th stage of the cell cycle. α is the goal of the number of BECs, and *k* is a constant. Assuming that the proliferation of HPCs also depends on a deficit of BECs, the proliferation ratio of HPCs at the *n*th stage can be described as$$c_{n}=la_{n}$$

Here, *l* is a constant that describes the proliferative ability of HPCs compared with that of BECs. If the differentiation ratio of HPCs is described as $b_{n}$ and the number of HPCs as $P_{n}$, the total dynamics of the cell number can be described as$$\left(\begin{array}{c} X_{n+1}\\ P_{n+1} \end{array}\right)=\left(\begin{array}{cc} 1+a_{n}-d & b_{n}\\ 0 & {1+c}_{n}-b_{n}-d \end{array}\right)\left(\begin{array}{c} X_{n}\\ P_{n} \end{array}\right)$$

The dynamic change in cell number based on the above mathematical model [with the settings of α = 300, *k* = 0.001, *l* = 3.5, $b_{n}$ = 0.5 (constant) and *d* = 0.1] was simulated (Appendix A). The number of BECs gradually increased with increasing differentiation of HPCs and then reached a plateau when the number of BECs increased. As part of the sensitivity analysis, cellular dynamics were analyzed with various coefficients (Appendix A). The results were generally similar, suggesting that the mathematical model proposed here is robust, and that small differences in coefficients between cell types, injury models, and time course of injury may not affect the results extensively.

### 3.7. dHPC-Dependent Regeneration of BECs in the Mathematical Model

Next, this mathematical model was applied to a mouse model of tdTomato-labeled BECs (Figure 6A). BECs were categorized into tdTomato-positive BECs (X) and tdTomato-negative BECs (Y), and similarly, HPCs were categorized into tdTomato-positive HPCs (P) and tdTomato-negative HPCs (Q). The coefficient “e” was used to describe the dedifferentiation ratio of BECs to HPCs. Note that X (P) and Y (Q) share a coefficient because they have the same characteristics except for the expression of tdTomato. The proliferative ratios of BECs and HPCs in this model are described by$$a_{n}=k\left(\alpha -{(X}_{n}+Y_{n})\right)$$$$c_{n}=la_{n}$$

The dedifferentiation ratio of BECs to HPCs ($e_{n})$ is assumed to depend on a deficiency of HPCs, which is described as follows:$$e_{n}=\varepsilon (\beta -{(P}_{n}+Q_{n}))$$

Here, β is the goal of the number of HPCs, and ε is the constant used to indicate the tendency of BECs to dedifferentiate into HPCs. In addition, differentiation ($b_{n}$) and dedifferentiation ($e_{n}$) should compete with each other, as described by$$b_{n}+e_{n}=m$$
where m is a constant that describes the ratio of cells whose differentiation status changes. Given these settings, the total dynamics of the cell number can be described by$$\left(\begin{array}{c} X_{n+1}\\ Y_{n+1}\\ P_{n+1}\\ Q_{n+1} \end{array}\right)=\left(\begin{array}{cccc} 1+a_{n}-e_{n}-d & 0 & b_{n} & 0\\ 0 & 1+a_{n}-e_{n}-d & 0 & b_{n}\\ e_{n} & 0 & {1+c}_{n}-b_{n}-d & 0\\ 0 & e_{n} & 0 & {1+c}_{n}-b_{n}-d \end{array}\right)\left(\begin{array}{c} X_{n}\\ Y_{n}\\ P_{n}\\ Q_{n} \end{array}\right)$$

The dynamic changes in cell numbers based on the above mathematical model [with settings of α = 300, *β* = 50, *k* = 0.001, *l* = 3.5, *m* = 0.5, and *d* = 0.1] were simulated (Figure 6B). The value of ε was set as three-staged to represent the KT (ε = 0.001), KT/TFF1^+/−^ (ε = 0.0025; 2.5-fold KT) and KT/TFF1^−/−^ (ε = 0.00625; 2.5-fold KT/TFF1^+/−^) mouse models. As expected, the number of tdTomato-labeled HPCs (dHPCs, described as *p*, red line) remained low in the KT model, whereas it increased in the KT/TFF1^−/−^ model at an early stage. The value of the coefficients in each cell cycle are shown in Appendix A. Naturally, differentiation of HPCs into BECs was lower and dedifferentiation of BECs into HPCs was higher in KT/TFF1KO mice; however, they tentatively reversed, and differentiation in KT/TFF1KO overcome KT when cell number of BECs increased (around cell cycle 10). These results indicate that the differentiation of dHPCs contributed to the regeneration of BECs paradoxically in the increased dedifferentiation model KT/TFF1^−/−^. In addition, the number of BECs and HPCs, as well as the ratio of tdTomato-positive BECs, after a cell cycle of 30 were similar to the observed data in the mouse model (Figure 6C–E), supporting the reproducibility of this mathematical model to represent the dHPC-dependent regeneration of BECs in the mouse model.

### 3.8. Many dHPC-Derived Hepatocytes Were Found in TFF1-Deficient Mice

Given that dHPCs differentiate into BECs, dHPCs might also have differentiated into hepatocytes. Thus, triple IF of CK19/tdTomato/HNF4α was performed to identify dHPC-derived hepatocytes. As expected, CK19^-^tdTomato^+^HNF4α^+^ cells (assumed to be dHPC-derived hepatocytes) were occasionally found (Figure 7A). The number of dHPC-derived hepatocytes was significantly greater in the KT/TFF1^−/−^ mice (Figure 7B), suggesting that loss of TFF1 resulted in the transdifferentiation of BECs into hepatocytes. To further evaluate the contribution of dHPC as supplier of hepatocytes, double IF of tdTomato/AFP (alpha-fetoprotein) was performed, showing double-positive cells with morphologically hepatocyte-like structure (Figure 7C). Quantification of the results (Figure 7D) revealed almost 10% of dHPC-derived hepatocytes was positive for AFP. Given that AFP works as a marker for newly generated hepatocytes, these results support the idea that regenerated hepatocytes are derived from dHPCs.

### 3.9. Different Mechanisms of BEC Regeneration in the Mouse Model of BDL and CCl_4_

To further examine bile duct regeneration in other models of liver injury, we used BDL and CCl_4_ treatment (Figure 8A,B). Double IF of CK19/tdTomato was performed (Figure 8C), the number of BECs was counted, and the positivity rate of tdTomato was calculated. Compared with that in the mice before treatment, the number of BECs in both models was slightly increased (approximately 1.5-fold greater). In the BDL model, the number of BECs did not differ among the genotypes, and the percentage of tdTomato-labeled cells was approximately 25%, without a significant difference among the mice (Figure 8D). Given that the percentage of tdTomato-BECs was lower than that in the mice before treatment, regenerated BECs were predominantly derived from the cells without the label of tdTomato in the BDL mouse model. Similarly, the number of BECs and the ratio of tdTomato-labeled BECs did not significantly differ among the mice in the CCl_4_ group; however, the tdTomato-positive rate was nearly 40%, which was equivalent to that of the mice before treatment (Figure 8E), suggesting that the regeneration of BECs depends on tdTomato-positive cells in the CCl_4_ model.

Given that TFF1 deficiency did not affect the proportion of tdTomato-labeled BECs in the BDL and CCl_4_ models, the dedifferentiation of BECs likely did not contribute to the regeneration of BECs in these models. As such, if the regeneration of BECs depends on HPCs, the ratio of tdTomato-positive BECs should decrease (corresponding to Figure 4C). In contrast, if the regeneration of BECs depends solely on the self-proliferation of mature BECs, the ratio of tdTomato-positive BECs should be maintained (corresponding to Figure 4B). The former hypothesis matches the results of the BDL model and the latter the results of the CCl_4_ model.

## 4. Discussion

In this study, we investigated biliary regeneration in KT/TFF1^−/−^ mice and several liver injury models. During CDE-induced liver injury, biliary regeneration is dependent on the differentiation of HPCs. Notably, frequent dedifferentiation of BECs to dHPCs was detected, and dHPC-derived BECs and hepatocytes were observed in KT/TFF1^−/−^ mice. These findings suggested that the loss of TFF1 promoted the dedifferentiation of BECs and, paradoxically, resulted in an increase in dHPC-derived BECs and hepatocytes.

The involvement of HPCs in the regeneration of hepatocytes has been discussed [27], whereas the association between HPCs and BECs has not been extensively investigated. Recent studies, however, revealed that BECs can transdifferentiate into hepatocytes when the proliferation of hepatocytes is inhibited by genetic intervention or severe injury [6,7]. Most recently, Pu et al. demonstrated the existence of transitional liver progenitor cells (TLPCs), which originate from BECs and differentiate into both hepatocytes and BECs [28]. These TLPCs presumably correspond to dHPCs in our mouse models. While these previous studies emphasized the transdifferentiation of BECs into hepatocytes, we focused on biliary regeneration and demonstrated that frequent dHPCs appeared and became the predominant provider of mature BECs in KT/TFF1^−/−^ mice. The fact that the differentiation of dHPCs is accelerated in the dedifferentiation model (KT/TFF1^−/−^) might be perplexing; however, our mathematical models indicated that dedifferentiation could cause the tentatively accelerated differentiation of dHPCs. Nevertheless, the number of BECs was significantly lower in the KT/TFF1^−/−^ mice than in the control mice, suggesting that the loss of TFF1 basically resulted in the dedifferentiation of BECs and inhibited their regeneration. In addition, we have previously reported that loss of TFF1 resulted in the increase in liver weight and the decrease in liver fibrosis after liver injury, suggesting that loss of TFF1 contributed to the regeneration of the liver probably due to the differentiation of dHPCs into hepatocytes [22].

TFF1 plays an important role in protecting and regenerating the mucous membranes of the gastrointestinal tract, such as the stomach and intestines. TFF1 is also expressed in the respiratory tract epithelium, pancreas and liver and is involved in tissue regeneration and repair in these tissues [19,20,24,25]. In this study, the relationship between HPCs and TFF1 was identified, especially in the CDE model, which is a well-known liver damage model that shows many HPCs [29]. While TFF1-associated dedifferentiation of BECs contributed to biliary regeneration in the CDE model, this process did not contribute to biliary regeneration in CCl_4_ and BDL models. Generally speaking, CDE diet induces apoptosis of hepatocytes across large areas of the liver, resulting in liver regeneration of high extent, thus requires high number of HPCs. On the other hand, CCl_4_ induces necrosis of hepatocytes around central vein, resulting in the activation of Kupffer cells and liver fibrosis, which probably does not require the aid of HPCs for biliary regeneration. BDL model induces apoptosis of cholangiocytes and hepatocytes around Glisson’s sheath with moderate regeneration of BECs, requiring the supply of BECs from HPCs. One can assume that high demand of BECs and/or hepatocytes cause expansion of HPCs thus results in the dedifferentiation of BECs to HPCs in CDE model. From the mechanistic perspective of TFF1-regulated differentiation, previous reports revealed that Wnt signaling promotes hepatic regeneration and that Notch signaling promotes biliary regeneration in a competitive manner [30]. We previously reported that loss of TFF1 was associated with Wnt/ Notch pathways in vivo; Notch signaling pathway in the liver was downregulated in TFF1KO mice after CDE diet [22] and β-catenin was activated in hepatocellular carcinoma found in TFF1KO mice [25]. In addition, overexpression of TFF1 inhibited Wnt signaling pathway in hepatocytes [25] and pancreatic cancer cells in vitro [31]. Thus, it is reasonable to assume that the loss of TFF1 results in the acceleration of Wnt signaling, the inhibition of Notch signaling, a small number of BECs, and eventually, frequent HPC-derived hepatocytes in mice. If this role of TFF1 is relevant in humans, treatment with TFF1 might be able to accelerate biliary regeneration. Patients with biliary dysfunction, such as primary biliary cholangitis and sclerosing cholangitis, suffer from the inflammation and destruction of bile ducts, thus TFF1-treatment with these patients can overcome their clinical conditions.

Recent single-cell studies have revealed the transcriptomic characterization of cholangiocytes and its plasticity. Interestingly, MacParland et al. identified Trefoil Factors as markers of mature BECs [32]. Given the findings of our study, TFF1 might not be just a marker of BECs but the actual player that regenerates BECs. In terms of HPCs, Segal et al. revealed the presence of TROP2-positive hepatobiliary hybrid progenitors in human fetal liver [33], which has been identified in mouse models including lineage-tracing studies [26]. On the other hand, Gribben et al. suggested that transdifferentiation events take place between cholangiocytes and hepatocytes without the presence of stem cells or progenitor cells in human chronic liver disease [34]. Although our study suggested that dHPCs mediate the transdifferentiation of cholangiocytes to hepatocytes, actual stemness or progenitor-like features of dHPCs (including the expression status of TROP-2, EpCAM, NCAM, Lgr5, etc.) remains to be clarified.

In conclusion, we found that biliary regeneration depends on the differentiation of HPCs in CDE-induced liver injury and that the loss of TFF1 resulted in the dedifferentiation of BECs to HPCs. The ability of TFF1 to accelerate the differentiation of HPCs into BECs might be useful for treating patients with biliary-related dysfunctions, such as primary biliary cholangitis (PBC) and primary sclerosing cholangitis (PSC). While the downstream molecular mechanisms of TFF1 are not fully elucidated, this study establishes a clear in vivo framework of dedifferentiation-based regeneration, which could serve as a foundation for future mechanistic investigations.

## Figures and Tables

**Figure 1 cells-14-01323-f001:**
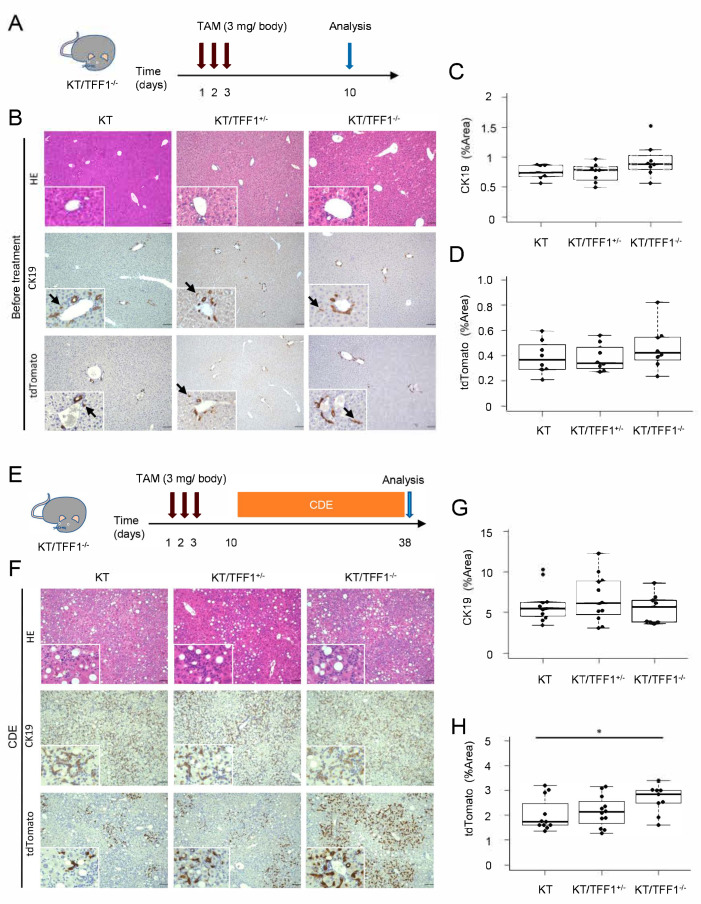
Loss of TFF1 resulted in the proliferation of tdTomato-labeled cells. (**A**) Scheme of mouse treatment with TAM. (**B**) Representative images of HE and IHC of the liver before treatment. The arrows indicate micro BECs. (**C**,**D**) Quantification of the areas positive for CK19 and tdTomato in the KT (*n* = 8), KT/TFF1^+/−^ (*n* = 8) and KT/TFF1^−/−^ mice (*n* = 8). (**E**) Scheme of mouse treatment with the CDE diet. (**F**) Representative images of HE and IHC staining of the liver after CDE treatment. (**G**,**H**) Quantification of the areas positive for CK19 and tdTomato in the KT (*n* = 11), KT/TFF1^+/−^ (*n* = 12) and KT/TFF1^−/−^ mice (*n* = 9). The data are presented as the means ± SEMs. Student’s *t* test, * *p* < 0.05. Scale bars: 100 μm.

**Figure 2 cells-14-01323-f002:**
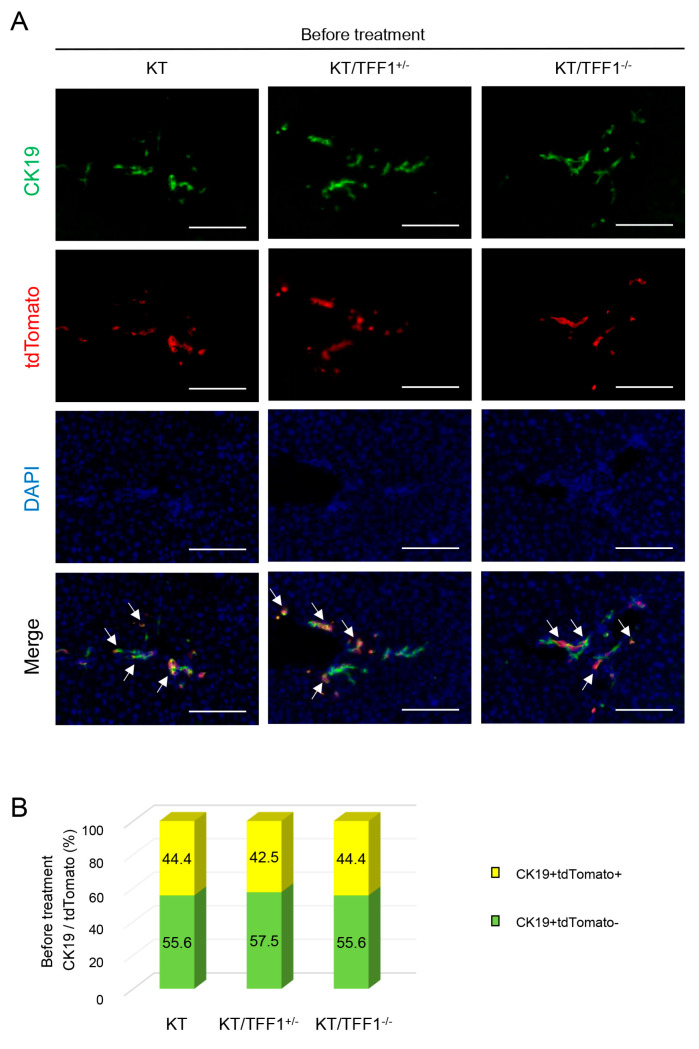
The labeling efficacy was equivalent among the mice. (**A**) Representative fluorescence images of CK19 and tdTomato before treatment. The arrows indicate tdTomato-labeled BECs. (**B**) Quantification of the tdTomato-labeled BECs in the KT (*n* = 8), KT/TFF1^+/−^ (*n* = 8) and KT/TFF1 ^−/−^ mice (*n* = 8) before treatment. Scale bars: 100 μm.

**Figure 3 cells-14-01323-f003:**
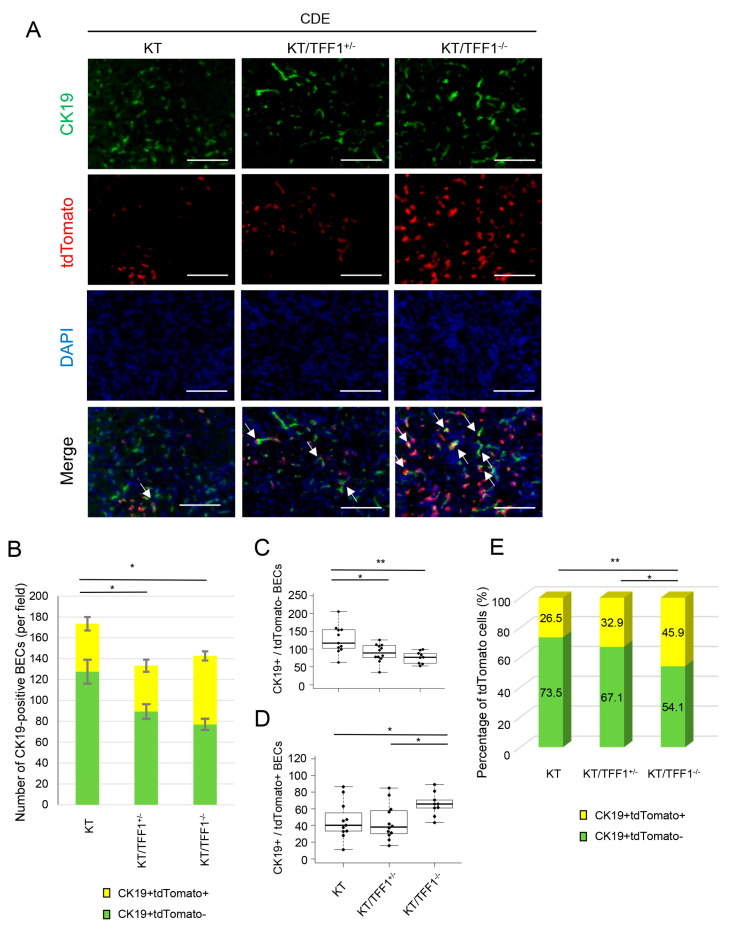
The proportion of tdTomato-positive BECs after CDE varied depending on the mouse model. (**A**) Representative fluorescence images of CK19 and tdTomato after CDE treatment. The arrows indicate tdTomato-labeled BECs. (**B**) Quantification of the number of CK19-positive BECs after CDE treatment in the KT (*n* = 11), KT/TFF1^+/−^ (*n* = 12) and KT/TFF1^−/−^ mice (*n* = 9). (**C**,**D**) Quantification of the number of tdTomato-labeled and non-labeled BECs in the KT (*n* = 11), KT/TFF1^+/−^ (*n* = 12) and KT/TFF1^−/−^ mice (*n* = 9). (**E**) The percentage of tdTomato-labeled BECs in the KT (*n* = 11), KT/TFF1^+/−^ (*n* = 12) and KT/TFF1^−/−^ mice (*n* = 9). The data are presented as the means ± SEMs. Student’s *t* test, * *p* < 0.05 and ** *p* < 0.01. Scale bars: 100 μm.

**Figure 4 cells-14-01323-f004:**
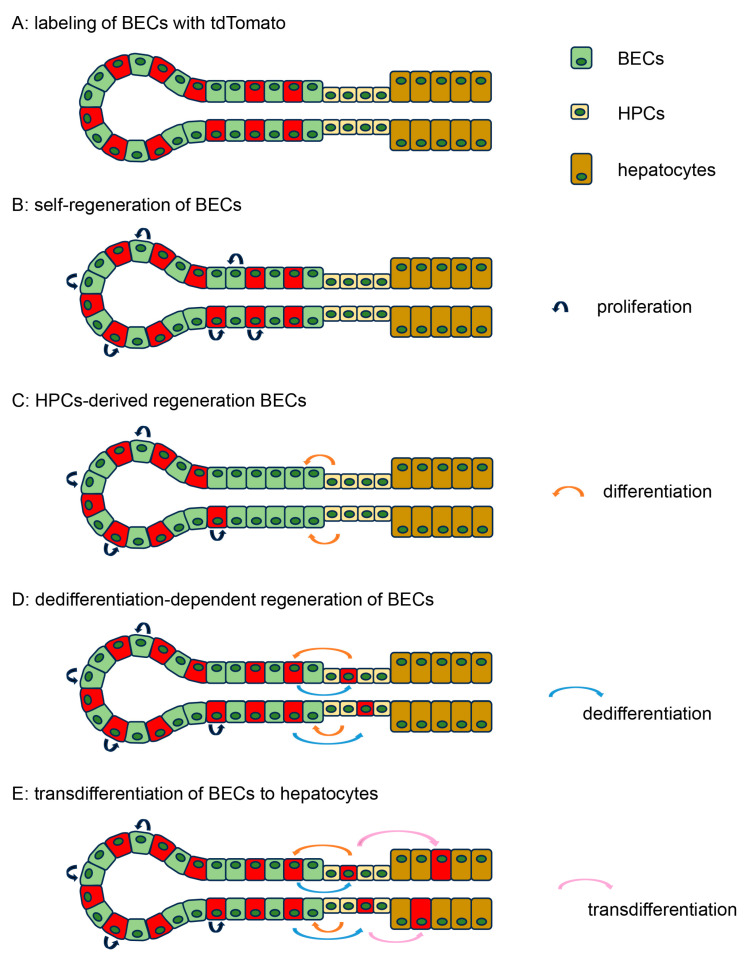
Model of the differentiation of HPCs and biliary regeneration. Schematic models of biliary regeneration with the differentiation of HPCs. (**A**) Labeling of BECs with tdTomato (before treatment), (**B**–**E**) distribution of tdTomato-labeled cells after liver damage depending on the regenerative mechanism ((**B**), self-duplication; (**C**), HPC dependent; (**D**), dedifferentiation dependent; (**E**), transdifferentiation).

**Figure 5 cells-14-01323-f005:**
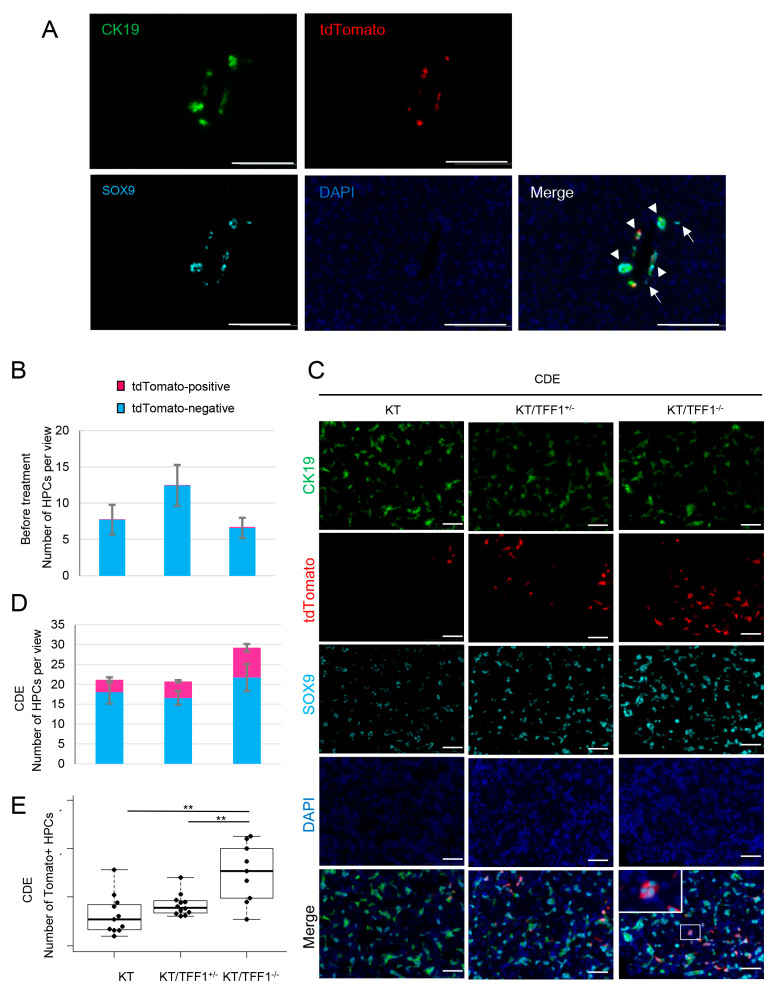
tdTomato-labeled BECs frequently dedifferentiate into HPCs in KT/TFF1^−/−^ mice. (**A**) Representative fluorescence images of CK19, tdTomato and SOX9 before treatment. The arrows indicate HPCs, and the arrowheads indicate mature BECs. Scale bars: 100 μm. (**B**) Quantification of the number of HPCs before treatment in the KT (*n* = 11), KT/TFF1^+/−^ (*n* = 12) and KT/TFF1^−/−^ mice (*n* = 9). (**C**) Representative fluorescence images of CK19, tdTomato and SOX9 after CDE treatment. Scale bars: 50 μm. (**D**,**E**) Quantification of the number of tdTomato-labeled HPCs in the KT (*n* = 11), KT/TFF1^+/−^ (*n* = 12) and KT/TFF1^−/−^ mice (*n* = 9). The data are presented as the means ± SEMs. Student’s *t* test, ** *p* < 0.01.

**Figure 6 cells-14-01323-f006:**
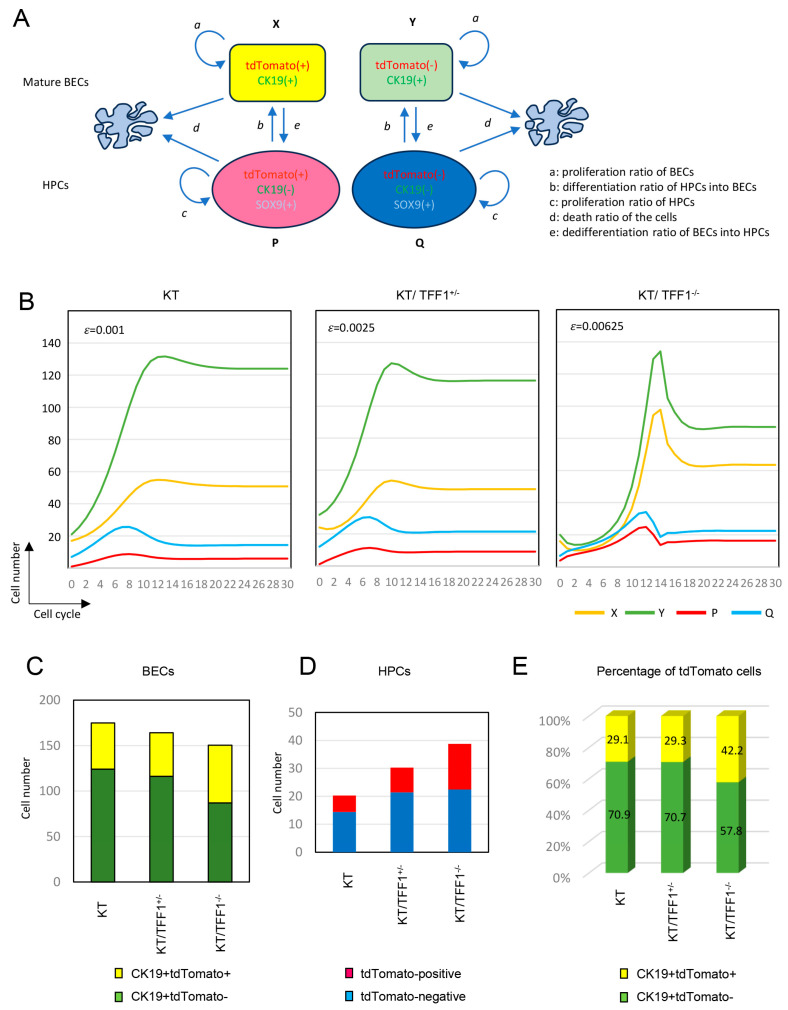
dHPC-dependent regeneration of BECs in the mathematical model. (**A**) Scheme of the mathematical model for dHPC-dependent regeneration of BECs. (**B**) Simulation of the dynamic change in the number of BECs and HPCs. (**C**–**E**) Cell numbers and percentages of tdTomato-labeled BECs and HPCs at the cell cycle of 30.

**Figure 7 cells-14-01323-f007:**
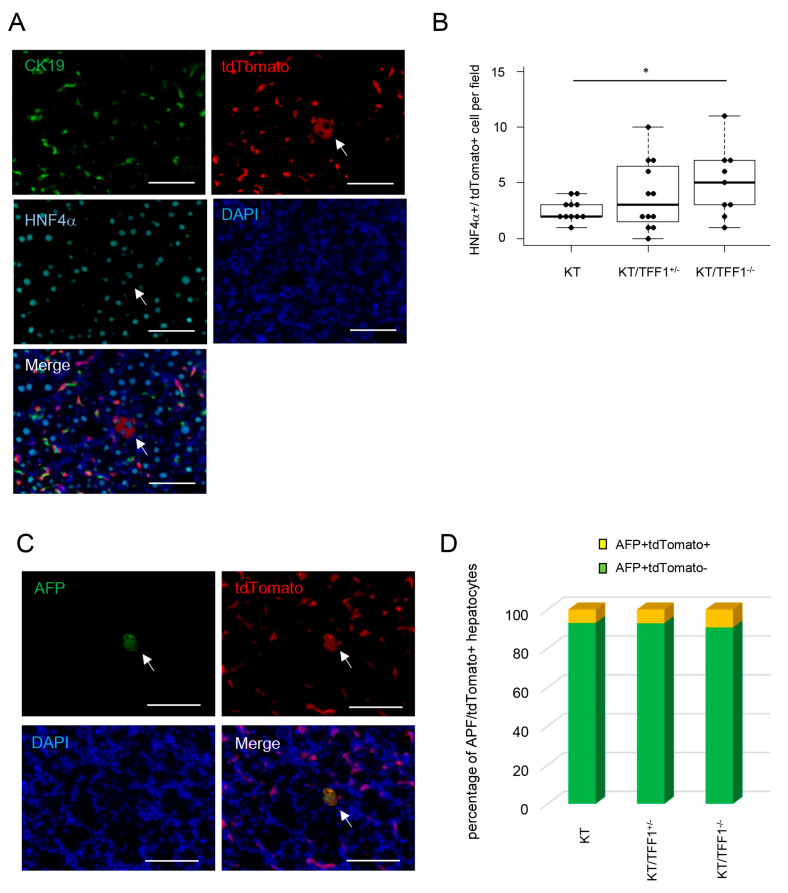
Frequent dHPC-derived hepatocytes were found in TFF1-deficient mice. (**A**) Representative fluorescence images of CK19, tdTomato and HNF4α after CDE treatment. (**B**) Quantification of the number of tdTomato-labeled hepatocytes in the KT (*n* = 11), KT/TFF1^+/−^ (*n* = 12) and KT/TFF1^−/−^ mice (*n* = 9). The data are presented as the means ± SEMs. (**C**) Representative fluorescence images of AFP and tdTomato after CDE treatment. (**D**) Quantification of the percentage of tdTomato/AFP-positive cells in the KT (*n* = 11), KT/TFF1^+/−^ (*n* = 12) and KT/TFF1^−/−^ mice (*n* = 9). Student’s *t* test, * *p* < 0.05. Scale bars: 100 μm.

**Figure 8 cells-14-01323-f008:**
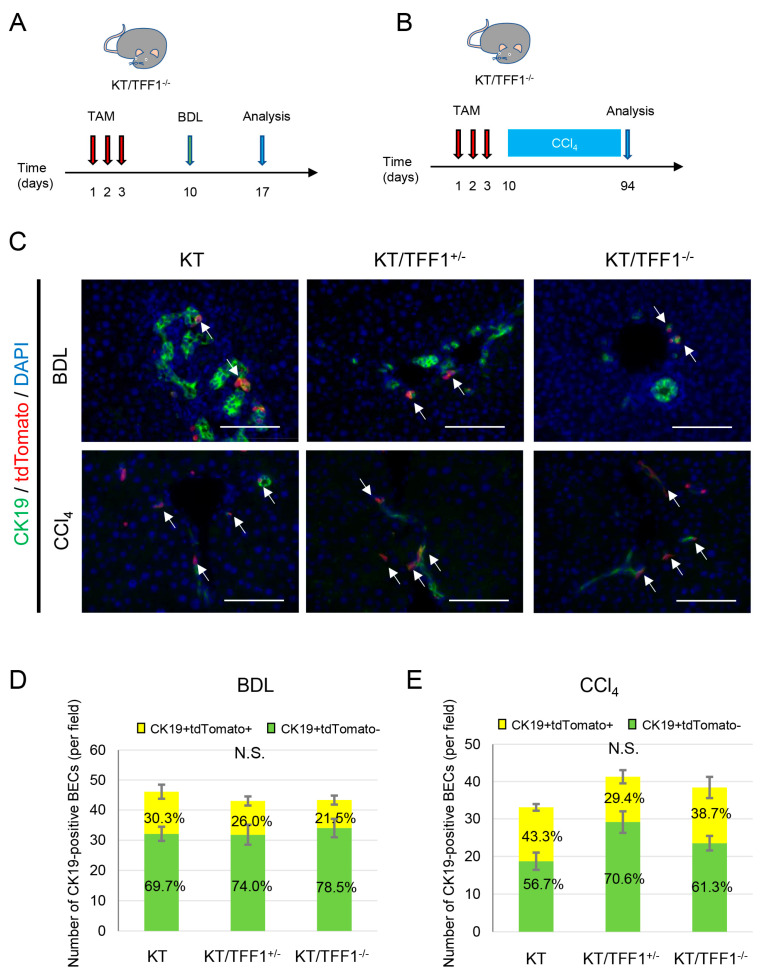
Different mechanisms of BEC regeneration in the mouse models of BDL and CCl_4_. (**A**) Scheme of mouse treatment by BDL. (**B**) Scheme of mouse treatment with CCl_4_. (**C**) Representative fluorescence images of double immunostaining of CK19 and tdTomato after BDL and CCl_4_ treatment. (**D**,**E**) Quantification of the number of tdTomato-labeled BECS after BDL and CCl_4_ treatment. The data are represented as the mean ± SEMs. Student’s *t* test, significance threshold was *p* < 0.05. Scale bars: 100 μm.

**Table 1 cells-14-01323-t001:** Conditions and primary antibodies used for immunohistochemistry and immunofluorescence.

Antigen	Company	Clone	Host	Concentration
CK19	Developmental Studies Hybridoma Bank (Iowa, IA, USA)	TROMA-III	rat	1:100
RFP	ROCKLAND (Limerick, PA, USA)	600-401-379	rabbit	1:1000
SOX9	EMD Millipore (Burlington, MA, USA)	AB5535	rabbit	1:500
HNF4a	Santa Cruz Biotechnology (Dallas, TX, USA)	sc-6556	goat	1:50
AFP	R and D Systems (Minneapolis, MN, USA)	AF5369	goat	1:100

## Data Availability

Datasets available on request from the authors.

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
