# Peer review of "Dedifferentiation-Dependent Regeneration of the Biliary Ductal Epithelium in Response to Hepatic Injury in TFF1-Deficient Mice"

_cells, 2025, doi:10.3390/cells14171323_

Round 1
Reviewer 1 Report
Comments and Suggestions for Authors
This manuscript investigates the mechanisms of biliary epithelial cell (BEC) regeneration following liver injury, focusing on the role of trefoil factor 1 (TFF1) using sophisticated lineage tracing in genetically engineered mouse models. The study provides novel insights into the dedifferentiation of BECs into hepatic progenitor cells (HPCs) and their subsequent contribution to biliary and hepatocyte regeneration, particularly under TFF1-deficient conditions. The work is timely, methodologically sound, and addresses a significant gap in the understanding of liver regeneration and epithelial plasticity.
Major Strengths
- The use of KRT19-creERT/LSL-tdTomato lineage tracing in combination with TFF1 knockout mice is a powerful strategy to dissect cell fate and regeneration dynamics in vivo.
- Multiple liver injury models (CDE diet, CCl4, BDL) are employed, allowing the authors to distinguish context-dependent mechanisms of regeneration.
- The inclusion of a mathematical model to simulate cell population dynamics adds depth and predictive value to the biological findings.
Major Concerns and Suggestions
- While the study demonstrates that TFF1 loss promotes BEC dedifferentiation into HPCs and alters regeneration dynamics, the downstream molecular mechanisms (e.g., Wnt/Notch signaling) are only briefly discussed and not directly interrogated. The authors should strengthen the mechanistic aspect by:Including additional data (if available) on the activation status of Wnt and Notch pathways in TFF1-deficient vs. control mice (e.g., β-catenin, Notch target gene expression). At minimum, expanding the discussion to more critically address how TFF1 may interact with these pathways, referencing their own and others’ previous work.
- The manuscript focuses on cellular lineage and population dynamics but provides limited information on the functional consequences of altered regeneration (e.g., biliary function, liver injury markers, fibrosis). If possible, include data on liver function tests (ALT, AST, bilirubin), biliary function, or fibrosis assessment (e.g., Sirius Red staining) in the different models and genotypes; Discuss how the observed cellular changes translate to organ-level outcomes and disease relevance.
- The mathematical model is a valuable addition, but its description is somewhat technical and may be difficult for some readers to follow. Provide a more intuitive explanation of the model’s assumptions and parameters in the main text or as a figure legend. Consider including a supplementary table summarizing parameter values and their biological meaning.
- The discussion could better integrate recent advances in the field, such as the identification of transitional liver progenitor cells (TLPCs) and the plasticity of cholangiocytes. Expand the discussion to compare and contrast your findings with recent single-cell and lineage tracing studies; Discuss the potential clinical implications for diseases such as primary biliary cholangitis and sclerosing cholangitis.
Minor Points
Clearly state the number of animals and replicates for each experiment in figure legends. Specify the statistical tests used and define significance thresholds. Ensure all abbreviations are defined at first use in the main text and figures.
Reviewer 2 Report
Comments and Suggestions for Authors
This manuscript by Yamamoto et al. explores the role of TFF1 in the dedifferentiation-dependent regeneration of biliary ductal epithelium following hepatic injury. The authors utilized lineage tracing with KRT19-creERT/LSL-tdTomato mice to demonstrate how TFF1 deficiency enhances the dedifferentiation of biliary epithelial cells (BECs) into hepatic progenitor cells (HPCs), subsequently driving their differentiation into hepatocytes and BECs in various liver injury models (CDE, CCl4, and BDL). The findings effectively support the hypothesis that TFF1 plays a critical role in biliary regeneration. Additionally, the authors developed a novel mathematical model to quantify and simulate the observed experimental phenomena, reinforcing the concept of dedifferentiation-based regeneration.
Issues and Concerns:
- The methods section is generally robust, but certain methodological clarifications are recommended, such as providing justification for the selected dose of tamoxifen (3 mg for 3 days) and the one-week washout period. Clarifying whether these parameters were optimized through preliminary experiments would enhance methodological transparency.
- The identification of HPCs as SOX9+CK19- cells warrants further clarification because SOX9 is also expressed in mature BECs. Using a more specific marker or combination of markers (e.g., EpCAM, AFP, CD133) could strengthen the validity of HPC identification. Additionally, clarification on whether quantification of tdTomato, CK19, and SOX9-positive cells was conducted in a blinded manner would help address potential bias.
- For the mathematical model, explicitly providing justification or performing sensitivity analyses for parameter choices (e.g., proliferation and differentiation rates) would significantly strengthen confidence in the model's robustness and biological relevance.
- The mathematical model assumes a constant cell death rate ("d"); however, actual liver injury models might exhibit different apoptosis or necrosis rates. Experimentally validating or clearly justifying this assumption would improve the model’s accuracy and applicability.
- Clarification regarding the intensity and severity differences among the hepatic injury models (CDE, BDL, and CCl4) is essential. Specifically, it would be beneficial to elaborate on why TFF1 deficiency prominently affects dedifferentiation primarily in the CDE model, as opposed to the other models.
- The observation of transdifferentiation into hepatocytes is intriguing. However, incorporating additional hepatocyte-specific markers such as Albumin or hepatocyte-specific CYP enzymes would further validate the identity and functional capacity of these hepatocyte-like cells derived from dedifferentiated HPCs.
Overall, the manuscript presents a well-structured and insightful study that significantly advances our understanding of hepatic epithelial regeneration mechanisms, emphasizing the crucial role of TFF1.
Reviewer 3 Report
Comments and Suggestions for Authors
This study investigates biliary epithelial cell (BEC) regeneration and the role of trefoil factor 1 (TFF1) in liver injury using lineage-tracing mouse models (KT and KT/TFF1KO). After tamoxifen-induced labeling of CK19-positive BECs, mice were subjected to liver injury via a choline-deficient, ethionine-supplemented (CDE) diet. Results showed that TFF1 deficiency promoted dedifferentiation of BECs into hepatic progenitor cells (HPCs), termed dHPCs, which subsequently contributed to BEC and hepatocyte regeneration. In KT/TFF1KO mice, tdTomato-labeled BECs persisted post-injury, while their proportion decreased in wild-type mice, suggesting TFF1 loss enhances dHPC-dependent regeneration. Mathematical modeling supported this mechanism, demonstrating that increased dedifferentiation (simulated by higher ε values in TFF1KO) paradoxically elevated tdTomato-positive BECs via dHPC differentiation. Additionally, TFF1 deficiency accelerated transdifferentiation of BECs into hepatocytes. Contrastingly, bile duct ligation (BDL) and CCl4 models revealed TFF1-independent regeneration pathways: BDL relied on tdTomato-negative cell proliferation, whereas CCl4 depended on tdTomato-positive BEC self-renewal. The study highlights TFF1’s critical role in regulating BEC plasticity, with its loss favoring dedifferentiation over direct proliferation. These findings suggest TFF1 as a potential therapeutic target for biliary disorders, though several concerns need to be addressed.
1.TFF1’s down steam gene expression alteration should be discussed and anlyzed.
2.Could these phenomenon validated in human related models? Such as organoids?
Round 2
Reviewer 2 Report
Comments and Suggestions for Authors
The authors have addressed most of the concerns raised by the reviewers for the original submission. The manuscript is suitable for publishing in its current format.
Reviewer 3 Report
Comments and Suggestions for Authors
All concerns has been addressed.